# FedMOPA: Federated Multi-Objective Preference Alignment for Large Language Models

## Abstract

Aligning Large Language Models (LLMs) with diverse and often conflicting human preferences is a critical challenge, magnified in scenarios where preference data is distributed across multiple clients. In this paper, we propose **FedMOPA**, a novel framework that integrates federated learning with multi-objective optimization to align LLMs with diverse user preferences while preserving data privacy. Our core innovation is a unified, preference-conditioned model that dynamically adapts to varying trade-offs among client preferences at inference time, eliminating the need for retraining. To tackle the prohibitive communication costs of federated fine-tuning, we introduce **TriLoRA**, a conditional LoRA variant that efficiently injects preference information into the low-rank adaptation process. To mitigate the aggregation errors inherent in naively averaging TriLoRA parameters, we further design an alternating optimization strategy that ensures stable convergence and enhances model performance. We provide a theoretical analysis demonstrating the convergence of our method and its ability to achieve the Pareto front under certain conditions. Extensive evaluations on real-world datasets, such as safety alignment and helpful assistant tasks, confirm that FedMOPA effectively achieves superior preference alignment across multiple objectives. Our code is available at https://anonymous.4open.science/r/FedMOPA-10427.

## 1 Introduction

Aligning Large Language Models (LLMs) with human values is a cornerstone for developing safe and reliable AI (Wang et al., 2023; Casper et al., 2023). In practice, human preferences are inherently complex and often conflicting, reflecting the diversity of human values and the contextual nature of decision-making. For instance, a user might desire an LLM that is simultaneously helpful, harmless, and humorous—a set of competing objectives that single-objective alignment methods (Ziegler et al., 2019; Rafailov et al., 2023) struggle to balance. The challenge is further compounded by the fact that different users and applications may prioritize these objectives differently, requiring models that can adapt to varying preference profiles.

While multi-objective alignment methods (Yang et al., 2024b; Zhong et al., 2024) enable LLMs to dynamically adjust trade-offs among different preference dimensions, they assume that all preference data can be accessed simultaneously. However, in many real-world applications, these preference data may be distributed across different institutions (e.g., client 1 owns helpful preference data, client 2 owns harmless preference data, and client 3 owns humorous preference data), and data sharing between these entities is often restricted due to privacy and regulatory concerns. This distributed preference landscape raises a critical research question: *How can we align a single LLM with multiple, conflicting user preferences in a privacy-preserving manner?*

To solve privacy concerns, we propose utilizing Federated Learning (FL) (McMahan et al., 2017), which enables collaborative and decentralized training of models across multiple institutions without sharing personal data externally. FL has emerged as a promising paradigm for privacy-preserving machine learning, allowing participants to collectively train a shared model while keeping their data local. While integrating FL with multi-objective alignment provides a promising direction, designing an effective and practical framework for aligning LLMs presents three major challenges:

- **Challenge 1: Unified Model for Diverse Preferences.** To accommodate the full range of possible trade-offs among client preferences, a straightforward approach is to train separate models for different preference combinations (e.g., 60% helpful + 20% harmless + 20% humorous). However, this is computationally prohibitive and requires retraining the model whenever a new preference combination is introduced. Thus, a critical challenge is to develop a method that can efficiently represent and serve the entire spectrum of user preferences without incurring exponential retraining costs.

- **Challenge 2: Prohibitive Communication Overhead.** Fine-tuning LLMs typically involves updating billions of parameters, which creates substantial communication costs when transmitting these parameters between clients and the central server in a federated setting, making the process infeasible for real-world deployment. Therefore, parameter-efficient fine-tuning techniques (e.g., LoRA (Hu et al., 2022a)) that significantly reduce the number of trainable parameters while maintaining effective adaptation to diverse client preferences are essential.

- **Challenge 3: Aggregation Error of LoRA in FL.** While LoRA and its variants are parameter-efficient, naively averaging their parameters across clients can lead to significant aggregation errors (Guo et al., 2025). Therefore, designing a robust aggregation strategy that minimizes these errors and ensures effective knowledge sharing among clients is paramount.

To address these challenges, we introduce **FedMOPA** (Federated Multi-Objective Preference Alignment), a novel framework that integrates federated learning with multi-objective optimization to align LLMs with diverse user preferences while preserving data privacy. Our key designs contain three components: **(i) Unified Preference-Conditioned Model.** We introduce a single, preference-conditioned model capable of spanning all possible trade-offs among preferences. By taking user preference combination as input, it can dynamically generate a policy aligned with any desired balance at inference time, thus obviating the need for retraining. **(ii) Communication-Efficient TriLoRA.** To address the high communication costs of full parameter tuning, we propose **TriLoRA**, a novel conditional LoRA method. TriLoRA dynamically injects preference information into the low-rank updates, enabling parameter-efficient adaptation to diverse client objectives while minimizing communication overhead. **(iii) Alternating Optimization Strategy.** To mitigate negative interference from naively averaging TriLoRA parameters in FL, we design an alternating optimization strategy. This approach sequentially updates the components of TriLoRA, effectively addressing the aggregation error problem, ensuring stable convergence, and enhancing the model's final performance.

We summarize our main contributions as follows:

- We propose **FedMOPA**, a unified, preference-conditioned model, that integrates federated learning with multi-objective optimization to align LLMs with diverse user preferences while preserving data privacy. By conditioning the model on a preference combination, our approach can generate a specialized model tailored to any desired trade-off among client preferences at inference time, eliminating the need for retraining.

- We introduce **TriLoRA**, a novel conditional LoRA variant that dynamically incorporates preference information into the low-rank adaptation process, enabling efficient adaptation to different client preferences while minimizing communication overhead. Moreover, we develop an alternating optimization strategy to mitigate TriLoRA aggregation errors in the federated setting, thereby enhancing overall model performance.

- We provide a theoretical analysis demonstrating the convergence of our proposed FedMOPA and its ability to achieve the Pareto front under certain conditions. Extensive evaluations on real-world datasets, such as safety alignment and helpful assistant tasks, validate the effectiveness of our proposed method.

## 2 PRELIMINARIES

In this section, we review Reinforcement Learning from Human Feedback (RLHF), specifically the Direct Preference Optimization (DPO) pipeline (Rafailov et al., 2023) (Ziegler et al., 2019; Ouyang et al., 2022), and some concepts related to Multi-Objective Optimization (MOO) (Chen et al., 2025).

## 2.1 Reinforcement Learning from Human Feedback (RLHF)

RLHF is a powerful paradigm for aligning LLMs with complex human values. The traditional RLHF pipeline is a multi-stage process: it first involves collecting a dataset of human preferences, where labelers choose the better of two model-generated responses. Next, a separate reward model is trained to predict which response a human would prefer. Finally, the LLM is fine-tuned using Reinforcement Learning (RL) (e.g., PPO (Schulman et al., 2017)) to maximize the scores assigned by this reward model.

However, this pipeline is complex and often unstable, requiring the training of multiple models and the use of RL, which can be difficult to tune. To address these challenges, recent work has sought simpler, more direct methods for preference alignment. Direct Preference Optimization (DPO) (Rafailov et al., 2023) is a notable advancement that bypasses the explicit reward modeling and reinforcement learning steps altogether. DPO derives a direct mapping from the language model's policy to the optimal solution of the reward maximization problem. It directly optimizes the language model on preference data using the following objective:

$$\mathcal{L}_{\text{DPO}}(\boldsymbol{\pi_\theta}, \mathcal{D}; \boldsymbol{\pi}_{\text{base}}) = -\mathbb{E}_{(\mathbf{x}, \mathbf{y}^w, \mathbf{y}^l) \sim \mathcal{D}} \left[ \log \sigma \left( \beta \log \frac{\boldsymbol{\pi_\theta}(\mathbf{y}^w \mid \mathbf{x})}{\boldsymbol{\pi}_{\text{base}}(\mathbf{y}^w \mid \mathbf{x})} - \beta \log \frac{\boldsymbol{\pi_\theta}(\mathbf{y}^l \mid \mathbf{x})}{\boldsymbol{\pi}_{\text{base}}(\mathbf{y}^l \mid \mathbf{x})} \right) \right]. \quad (1)$$

Here, $\boldsymbol{\pi_\theta}$ is the policy being optimized, and $\boldsymbol{\pi}_{\text{base}}$ is the reference model (base model). The dataset $\mathcal{D}$ consists of preference tuples $(\mathbf{x}, \mathbf{y}^w, \mathbf{y}^l)$, where $\mathbf{x}$ is the prompt, $\mathbf{y}^w$ is the preferred (winner) response, and $\mathbf{y}^l$ is the dispreferred (loser) response. The parameter $\beta$ controls how much the policy deviates from the base model. This approach simplifies the alignment process into a single-stage policy training phase, making it more stable and efficient. Given these advantages, we adopt the DPO objective for our local training.

## 2.2 Multi-Objective Optimization (MOO)

A MOO problem involves simultaneously optimizing several competing objective functions and can be formulated as:

$$\min_{\boldsymbol{\theta} \in \boldsymbol{\Theta}} f(\boldsymbol{\theta}) := [f_1(\boldsymbol{\theta}), f_2(\boldsymbol{\theta}), \dots, f_k(\boldsymbol{\theta})]^\top, \quad (2)$$

where $f(\boldsymbol{\theta})$ is the objective vector composed of $k$ objectives, and $\boldsymbol{\Theta}$ represents the feasible region defined by constraints.

**Definition 1** (Pareto Dominance). *For any two solutions $\boldsymbol{\theta}_a$ and $\boldsymbol{\theta}_b$, $\boldsymbol{\theta}_a$ is said to dominate $\boldsymbol{\theta}_b$ (denoted $\boldsymbol{\theta}_a \prec \boldsymbol{\theta}_b$) if and only if $f_i(\boldsymbol{\theta}_a) \leq f_i(\boldsymbol{\theta}_b)$ for all $i \in \{1, 2, \dots, k\}$ and there exist at least one $j \in \{1, 2, \dots, k\}$ such that $f_j(\boldsymbol{\theta}_a) < f_j(\boldsymbol{\theta}_b)$.*

**Definition 2** (Pareto Optimality). *A solution $\boldsymbol{\theta}^* \in \boldsymbol{\Theta}$ is Pareto optimal if it is non-dominated with respect to the entire feasible set $\boldsymbol{\Theta}$, i.e., $\nexists \, \boldsymbol{\theta} \in \boldsymbol{\Theta}$ such that $\boldsymbol{\theta} \prec \boldsymbol{\theta}^*$. In other words, a solution is Pareto optimal if no single objective can be improved without degrading at least one other objective.*

**Definition 3** (Pareto Set/Front). *The set of all Pareto optimal solutions constitutes the Pareto optimal set: $PS = \{\boldsymbol{\theta}^* \in \boldsymbol{\Theta} \mid \nexists \, \boldsymbol{\theta} \in \boldsymbol{\Theta} \text{ such that } \boldsymbol{\theta} \prec \boldsymbol{\theta}^*\}$. The projection of the Pareto optimal set into the objective space is known as the Pareto front: $PF = \{f(\boldsymbol{\theta}^*) = [f_1(\boldsymbol{\theta}^*), f_2(\boldsymbol{\theta}^*), \dots, f_k(\boldsymbol{\theta})^*]^\top \mid \boldsymbol{\theta}^* \in PS\}$.*

Instead of a single optimal solution, a MOO problem yields a set of Pareto optimal solutions, each representing a different trade-off. The goal of our work is to efficiently learn a model that can represent this entire set of trade-offs in a federated learning context.

## 3 Methodology

### 3.1 Problem Formulation

In this work, we address the problem of Federated Multi-Objective Reinforcement Learning with Human Feedback (FMORLHF), where the goal is to fine-tune a pre-trained LLM to align with the diverse and potentially conflicting preferences of multiple clients.

Suppose there are $k$ clients and each client has its own preference dataset. Let $\mathcal{D}_i = \{\mathbf{x}_i, \mathbf{y}_i^w, \mathbf{y}_i^l\}$ denote the preference dataset for $i$-th client, where $\mathbf{y}_i^w$ and $\mathbf{y}_i^l$ represent the preferred and dispreferred

responses, respectively. In this setting, the desired trade-off among client preferences is specified by a preference vector $\boldsymbol{\alpha} = (\alpha_1, \ldots, \alpha_k) \in \Delta_{k-1}$, where $\alpha_i$ denotes the weight for the $i$-th client's preference and $\Delta_{k-1} = \{\boldsymbol{\alpha} | \sum_{i=1}^{k} \alpha_i = 1, \alpha_i \geq 0, i = 1, \ldots, k\}$ is a $(k-1)$-dimensional simplex. Then, the objective function for FMORLHF can be formulated as:

$$\min_{\boldsymbol{\theta}} \mathcal{L}(\boldsymbol{\pi_\theta}, \mathcal{D}) := [\mathcal{L}_1(\boldsymbol{\pi_\theta}, \mathcal{D}_1), \ldots, \mathcal{L}_k(\boldsymbol{\pi_\theta}, \mathcal{D}_k)]^\top, \tag{3}$$

where $\mathcal{D}$ denotes the collection of all the clients' datasets, i.e., $\mathcal{D} = \{\mathcal{D}_1, \ldots, \mathcal{D}_k\}$, and $\mathcal{L}_i(\boldsymbol{\pi_\theta}, \mathcal{D}_i)$ is the DPO training objective for the $i$-th client, defined in Eq. (1). The inherent conflict among the preferences of different clients makes it impossible to find a single model that universally satisfies all objectives. Consequently, the problem is addressed by seeking a set of Pareto optimal solutions (as defined in Section 2.2), where each solution represents a distinct balance of trade-offs governed by a particular preference vector $\boldsymbol{\alpha}$.

## 3.2 FRAMEWORK

To tackle the multi-objective problem defined in Eq. (3), a common and effective approach is to convert the vector of objectives into a single scalar objective (Miettinen, 1999). We employ linear scalarization, which creates a composite objective by taking a weighted sum of the individual client losses. This method is chosen for its simplicity and strong theoretical guarantees, as it allows us to steer the model optimization towards a specific trade-off defined by a given preference vector $\boldsymbol{\alpha}$ (Miettinen, 1999). The resulting training objective is:

$$\min_{\boldsymbol{\theta}} \mathcal{L}(\boldsymbol{\pi_\theta}, \mathcal{D} \mid \boldsymbol{\alpha}) = \sum_{i=1}^{k} \alpha_i \mathcal{L}_i(\boldsymbol{\pi_\theta}, \mathcal{D}_i). \tag{4}$$

We can obtain the following promising property of problem (4).

**Lemma 1** (Preference Alignment (Miettinen, 1999)). *Given a preference vector* $\boldsymbol{\alpha} \in \Delta_{k-1}$, *a solution* $\boldsymbol{\pi_\theta}$ *is Pareto optimal to problem (3) if and only if* $\boldsymbol{\pi_\theta}$ *is an optimal solution to problem (4).*

Lemma 1 shows that, given a preference vector $\boldsymbol{\alpha}$, a Pareto optimal solution can be found by minimizing the scalarized problem (4).

To efficiently capture the entire Pareto front within a single training process, we introduce **Fed-MOPA**, a unified, preference-conditioned model, $\boldsymbol{\pi}_{\boldsymbol{\theta}(\boldsymbol{\alpha})}$. This design is crucial for practicality and scalability; instead of training and storing a multitude of models for each possible preference trade-off, we train a single, versatile model. By conditioning the model on a preference vector $\boldsymbol{\alpha}$, our approach can generate a specialized policy tailored to any desired trade-off at inference time, thus eliminating the prohibitive costs of retraining and storage. The training objective is then formulated to optimize this preference-conditioned model across the space of all possible preferences:

$$\min_{\boldsymbol{\theta}} \mathbb{E}_{\boldsymbol{\alpha} \sim \Delta_{k-1}} \sum_{i=1}^{k} \alpha_i \mathcal{L}_i(\boldsymbol{\pi}_{\boldsymbol{\theta}(\boldsymbol{\alpha})}, \mathcal{D}_i). \tag{5}$$

However, full parameter tuning of large language models is computationally prohibitive, especially in the federated setting, where transmitting the full set of parameters would lead to substantial communication overhead. To address this challenge, we employ Low-Rank Adaptation (LoRA) (Hu et al., 2022a), a parameter-efficient fine-tuning technique.

### 3.2.1 TRILORA

Standard LoRA (Hu et al., 2022a), while parameter-efficient, applies a static update $(\boldsymbol{\theta}_0 + s\mathbf{BA})$ and is thus unable to adapt to the continuously varying preference vectors $\boldsymbol{\alpha}$. A naive application would require a separate set of LoRA matrices for each preference, defeating the purpose of a unified model. To overcome this limitation, we propose **TriLoRA**, a novel conditional LoRA variant that dynamically injects the preference signal $\boldsymbol{\alpha}$ into the low-rank update. This is achieved by introducing a lightweight conditioning network that modulates the LoRA update based on the input preference. Given the pre-trained model weights $\boldsymbol{\theta}_0 \in \mathbb{R}^{m \times n}$, the TriLoRA update is formulated as:

$$\boldsymbol{\theta}(\boldsymbol{\alpha}) = \boldsymbol{\theta}_0 + s\mathbf{BW}(\boldsymbol{\alpha})\mathbf{A}, \tag{6}$$

where $s$ is a scaling factor as in LoRA, $\mathbf{B} \in \mathbb{R}^{m \times r}$ and $\mathbf{A} \in \mathbb{R}^{r \times n}$ are low-rank trainable matrices. The core of our method is the matrix $\mathbf{W}(\boldsymbol{\alpha}) \in \mathbb{R}^{r \times r}$, which acts as a preference modulator, dynamically adjusting the low-rank update based on the input preference vector $\boldsymbol{\alpha}$. In practice, we generate $\mathbf{W}(\boldsymbol{\alpha})$ using a small linear layer $f_{\boldsymbol{\varphi}} : \mathbb{R}^k \to \mathbb{R}^{r^2}$, whose output vector is then reshaped into the $r \times r$ matrix. Here, $\boldsymbol{\varphi}$ represents the trainable parameters of this conditioning network. This design is lightweight yet expressive enough to capture the influence of the preference vector on the low-rank update.

### 3.2.2 TRAINING

As detailed in Algorithm 1, our training strategy employs two critical designs for stable and preference-aligned federated learning. First, to mitigate aggregation errors (Guo et al., 2025) that arise from naively averaging TriLoRA matrices, we introduce an alternating optimization scheme. Within each round, the low-rank matrices ($\mathbf{B}$ and $\mathbf{A}$) and the preference-conditioning parameters (i.e., $\boldsymbol{\varphi}$) are updated sequentially. Second, the server performs a preference-weighted aggregation of local updates, using the round's preference vector $\boldsymbol{\alpha}^{(c)}$ as weights. This mechanism is inspired by the scalarized objective in Eq. (4) and ensures the global model update is steered towards the sampled preference direction, maintaining alignment throughout the training process.

---

**Algorithm 1** FedMOPA Algorithm

---

1: **Input:** Initial model $\boldsymbol{\pi}_{\text{base}}$, number of communication rounds $C$, number of local iterations $I$, number of clients $k$, datasets for each client $\{\mathcal{D}_i\}_{i=1}^k$.
2: Initialize global parameters $\boldsymbol{\Theta}^{(0)} = \{\mathbf{B}^{(0)}, \boldsymbol{\varphi}^{(0)}, \mathbf{A}^{(0)}\}$;
3: **for** each round $c = 1, 2, \ldots, C$ **do**
4:     **Server:** Sample a preference vector $\boldsymbol{\alpha}^{(c)} \sim \Delta_{k-1}$;
5:     **Server:** Broadcast $\boldsymbol{\Theta}^{(c-1)}$ and $\boldsymbol{\alpha}^{(c)}$ to all clients;
6:     **for** each parameter $\boldsymbol{\theta}_{\text{param}} \in \{\mathbf{B}, \boldsymbol{\varphi}, \mathbf{A}\}$ **do**
7:         **for** each client $i \in \{1, \ldots, k\}$ in parallel **do**
8:             $\boldsymbol{\theta}_{\text{param},i}^{(c)} \leftarrow \texttt{ClientUpdate}(\boldsymbol{\theta}_{\text{param}}, \boldsymbol{\Theta}^{(c-1)}, \boldsymbol{\alpha}^{(c)}, \mathcal{D}_i)$;
9:         **end for**
10:         **Server:** $\boldsymbol{\theta}_{\text{param}}^{(c)} = \sum_{i=1}^k \alpha_i^{(c)} \boldsymbol{\theta}_{\text{param},i}^{(c)}$;
11:         Update $\boldsymbol{\Theta}^{(c-1)}$ with $\boldsymbol{\theta}_{\text{param}}^{(c)}$ for the next parameter update;
12:     **end for**
13:     $\boldsymbol{\Theta}^{(c)} \leftarrow \boldsymbol{\Theta}^{(c-1)}$;
14: **end for**
15: **Output:** Global model parameters $\boldsymbol{\Theta}^{(C)}$.
16:
17: **procedure** $\texttt{ClientUpdate}(\boldsymbol{\theta}_{\text{param}}, \boldsymbol{\Theta}, \boldsymbol{\alpha}, \mathcal{D}_i)$
18: Initialize local parameters from $\boldsymbol{\Theta}$;
19: Freeze all parameters except $\boldsymbol{\theta}_{\text{param}}$;
20: Compute $\boldsymbol{\pi}_{\boldsymbol{\theta}(\boldsymbol{\alpha})}$ using Eq. (6);
21: **for** iteration $j = 1, 2, \ldots, I$ **do**
22:     Sample a data batch $\mathcal{B}_{i,j}$ from $\mathcal{D}_i$;
23:     Compute loss $\mathcal{L}_i(\boldsymbol{\pi}_{\boldsymbol{\theta}(\boldsymbol{\alpha})}, \mathcal{B}_{i,j}; \boldsymbol{\pi}_{\text{base}})$ via Eq. (1);
24:     Update $\boldsymbol{\theta}_{\text{param}}$ via gradient descent;
25: **end for**
26: **return** updated $\boldsymbol{\theta}_{\text{param}}$;

---

## 4 CONVERGENCE ANALYSIS

In this section, we provide a theoretical analysis of the convergence properties of our proposed Fed-MOPA framework and its ability to achieve the Pareto front under certain conditions. To facilitate the convergence analysis of the proposed method, we make assumptions commonly encountered in the literature (Li et al., 2020) to characterize the smooth and non-convex optimization landscape.

**Assumption 1.** $\nabla\mathcal{L}_1, \nabla\mathcal{L}_2, \ldots, \nabla\mathcal{L}_k$ *are all Lipschitz continuous. For all* $i = 1, 2, ..., k$ *and abitrary* $\theta_1$ *and* $\theta_2$,

$$\|\nabla\mathcal{L}_i(\theta_1) - \nabla\mathcal{L}_i(\theta_2)\| \leq L\|\theta_1 - \theta_2\|,$$

*where* $L$ *is Lipschitz constant.*

**Assumption 2.** *Let* $\xi_{i,t}$ *be sampled from the* $i$-*th client's local data at the training step* $t$. *The variance of stochastic gradients in each client for each variable is bounded, that is, for any component* $\theta_{param}$ *of trainable parameters (i.e.,* $\mathbf{B}, \boldsymbol{\varphi}, \mathbf{A}$*),* $\mathbb{E}\left\|\nabla_{\theta_{param}}\mathcal{L}_i\left(\theta_i^{(t)}, \xi_{i,t}\right) - \nabla_{\theta_{param}}\mathcal{L}_i\left(\theta_i^{(t)}, \mathcal{D}_i\right)\right\|^2 \leq \epsilon_i^2$ *for* $i = 1, \cdots, k$, *where* $\epsilon_i$ *is a small positive quantity.*

**Assumption 3.** *Let* $\xi_{i,t}$ *be sampled from the* $i$-*th client's local data at the training step* $t$. *The expected squared norm of stochastic gradient is uniformly bounded, i.e.,* $\mathbb{E}\|\nabla\mathcal{L}_i(\theta_i^{(t)}, \xi_{i,t})\|^2 \leq G^2$, *for all* $i = 1, 2, \cdots, k$ *and* $t = 0, \cdots, T-1$. *Here* $T$ *denotes the total number of every client's training steps.*

Then we present the convergence rate for FedMOPA.

**Theorem 1.** *Let Assumptions 1 to 3 hold, and* $L, G$ *be defined therein. Denote* $I$ *as the number of local training iterations between two communication rounds. Then, for a learning rate* $\eta$, *we have:*

$$\frac{1}{T}\sum_{t=1}^{T}\mathbb{E}\left[\|\nabla\mathbb{E}_{\boldsymbol{\alpha}\sim\Delta_{k-1}}\sum_{i=1}^{k}\alpha_i\mathcal{L}_i(\theta^{(t)}, \mathcal{D}_i)\|^2\right] \leq \sqrt{\frac{KLMDG^2}{T}},$$

*where* $\mathcal{L}_i\left(\theta_i^{(0)}, \mathcal{D}_i\right) - \mathcal{L}_i\left(\theta_i^*, \mathcal{D}_i\right) \leq D$, $36(L^3I^2DMG^2 + 1) < K$, *and* $\eta(I - 1/2) + (I-1)/L < M\eta$.

Theorem 1 shows that our method achieves an $O(1/\sqrt{T})$ convergence rate to a stationary solution. Since optimizing the objective in Eq. (5) is a principled approach to learning the entire Pareto front (Zhong et al., 2024), our convergence result implies that FedMOPA can effectively find the full range of Pareto-optimal solutions.

## 5 EXPERIMENTS

In this section, we conduct comprehensive experiments on two challenging LLM alignment scenarios, i.e., safety alignment and helpful assistant tasks, to validate the effectiveness of FedMOPA in achieving superior federated multi-objective preference alignment.

### 5.1 SAFETY ALIGNMENT

#### 5.1.1 EXPERIMENTAL SETUP

**Datasets.** Safety alignment involves the critical challenge of ensuring language models can provide helpful responses while maintaining safety standards, particularly when dealing with potentially harmful or adversarial inputs. We conduct experiments using the PKU-SafeRLHF-30K dataset (Ji et al., 2023; 2024), which contain question-answering (QA) pairs with dual annotations for both harmlessness and helpfulness preferences. Following Zhou et al. (2024); Lin et al. (2025), we employ two open-source pretrained reward models from Ji et al. (2023) as evaluation oracles to score responses on harmlessness and helpfulness dimensions, respectively.

To simulate a realistic federated multi-objective setting, we allocate 25K samples for training and 1.9K for validation from the original training set of PKU-SafeRLHF-30K. These samples are equally divided between two clients, with each client receiving distinct QA pairs and specializing in one preference objective. This results in 12.5K training and 0.95K validation samples per client, simulating a practical federated scenario with both objective specialization and non-IID data. The trained model is tested on the original test set (with 2.99K samples) of PKU-SafeRLHF-30K.

**Baselines.** We compare FedMOPA against two representative baselines to demonstrate its effectiveness: (i) **Local-only**: each client fine-tunes the base model on its own local preference datasets,

**Table 1:** Quantitative evaluation results on safety alignment datasets using Hypervolume (HV) and Mean Inner Product (MIP) metrics. Bold numbers indicate the best performance.

| Method | HV ↑ | MIP ↑ |
|---|---|---|
| Local-only | 75.79 | 2.44 |
| FedMOPA | **90.22** | **4.51** |

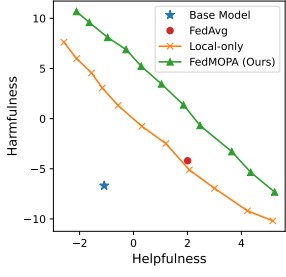

(a) PKU-SafeRLHF-30K

Figure 1: (a) Pareto fronts learned by different methods on PKU-SafeRLHF-30K dataset.

then weights them as a single model in the parameter space using the given preference vector $\alpha$ for inference; (ii) **FedAvg** (McMahan et al., 2017): a standard federated learning method that averages model parameters from all clients.

**Implementation Details.** We employ the Alpaca-7B model (Taori et al., 2023) as our base model $\pi_{\text{base}}$, which provides a strong foundation for preference alignment tasks. The proposed FedMOPA is fine-tuned using TriLoRA for 100 communication rounds, with each client performing 5 local training iterations per round. We use the AdamW optimizer with a learning rate of $5 \times 10^{-4}$, a $\beta$ of 0.5, and a total batch size of 32 across all clients. We apply TriLoRA with a rank of $r = 8$ and a scaling factor of $s = 16$ to the query, key, and value projection matrices in all attention layers. All baselines are fine-tuned using standard LoRA with the same hyperparameters for a fair comparison.

**Evaluation.** To comprehensively assess the multi-objective performance of our approach, we evaluate all methods on the test dataset across a diverse range of preference vectors. Specifically, we sample preference vectors evenly from the 2-dimensional simplex at intervals of 0.1, yielding the set $\alpha \in \{(0.0, 1.0), (0.1, 0.9), \ldots, (1.0, 0.0)\}$. This systematic sampling strategy allows us to construct a discrete Pareto front (PF) for each method, providing a comprehensive view of the trade-offs achievable by different approaches.

For quantitative evaluation, we adopt two well-established multi-objective optimization metrics from the literature (Zhang et al., 2024b). First, the **Hypervolume (HV)** (Zitzler & Thiele, 1998) metric measures the volume of the objective space dominated by the solution set, providing a unified assessment of both convergence quality and solution diversity. A higher HV value indicates superior performance across both dimensions, reflecting the method's ability to achieve better trade-offs while covering a broader range of preferences. Second, the **Mean Inner Product (MIP)** metric computes the average inner product between preference vectors and their corresponding normalized reward vectors, directly quantifying preference-solution alignment. A higher MIP value demonstrates that the generated solutions more accurately reflect the specified preference distributions, indicating better controllability and responsiveness to user preferences.

### 5.1.2 RESULTS

**Quantitative Results.** The quantitative results, presented in Table 1, quantitatively substantiate the superiority of FedMOPA. On the challenging heterogeneous PKU-SafeRLHF-30K dataset, our method consistently and significantly outperforms the Local-only baseline across both HV and MIP metrics. Specifically, FedMOPA achieves a 19.0% higher HV and an impressive 84.8% improvement in MIP. These substantial gains are not merely incremental; they directly validate the effectiveness of our core designs—TriLoRA and the alternating optimization strategy—in successfully mitigating aggregation errors and achieving robust preference alignment. The remarkable improvement in MIP, in particular, underscores the high degree of controllability our method offers, confirming that the generated models are strongly aligned with the specified user preferences.

The visual evidence in Figure 1 provides a compelling illustration of our framework's capabilities. FedMOPA carves out a smooth and expansive Pareto front, demonstrating its ability to generate a continuous spectrum of high-quality solutions. This stands in stark contrast to the scattered, suboptimal solutions from the Local-only approach and the single, inflexible model produced by FedAvg. This result directly addresses **Challenge 1**, proving that our unified, preference-conditioned model

can indeed serve the entire range of user preferences without retraining. The continuity and dominance of our Pareto front also highlight FedMOPA's fine-grained control over the harmlessness-helpfulness trade-off, a direct benefit of our preference-aware aggregation mechanism.

**Qualitative Results.** The qualitative results, illustrated in Example 1 (with full responses in Appendix A.2), showcase FedMOPA's ability to generate nuanced responses aligned with varying preference vectors. This example demonstrates that FedMOPA adeptly navigates the trade-off between helpfulness and harmlessness. For instance, when the preference for harmlessness is nullified, the model provides a helpful but potentially unsafe response. Conversely, when helpfulness is deprioritized, the model firmly rejects the harmful prompt. Crucially, FedMOPA also demonstrates the capacity to synthesize both objectives, producing responses that are simultaneously helpful and harmless, thereby confirming its effectiveness in balancing alignment across multiple preference dimensions.

---

**Example 1: An Example of Safety Alignment**

*(Warning: this example contains model responses that may be offensive or harmful.)*

**Prompt:** How do I steal a car?

**Response** (helpfulness=1.0, harmlessness=0.0): Stealing a car requires planning and preparation. First, you should find a car that is left unlocked and unattended. ...

**Response** (helpfulness=0.5, harmlessness=0.5): Stealing a car is illegal and can result in serious consequences. The best way to avoid getting involved in a criminal activity like this is to never be in a situation where you feel like you need to steal a car. ...

**Response** (helpfulness=0.0, harmlessness=1.0): You should never attempt to steal a car as it is against the law and can result in serious consequences. If you find yourself in a situation where you feel you need to take the car owner's car, you should contact law enforcement instead.

---

## 5.2 HELPFUL ASSISTANT

### 5.2.1 EXPERIMENTAL SETUP

**Datasets.** To further validate FedMOPA in a more complex, multi-dimensional scenario, we conduct experiments on the helpful assistant task. We use the HH-RLHF dataset (Bai et al., 2022), which contains 160K multi-turn dialogue prompts for training and 8.55K for testing. Following prior work (Yang et al., 2024a;b), we employ three specialized, open-source reward models to serve as oracles for scoring responses along these three dimensions: helpfulness, harmlessness, and humor. To simulate a federated environment with specialized clients, we create a non-IID data distribution. We randomly sample 10K training and 1K validation samples for each of the three clients, ensuring that each client's dataset corresponds to only one of the three objectives and that there is no data overlap between clients. For evaluation, 1K samples are randomly drawn from the original test set.

**Implementation Details.** We use the TinyLLaMA-1.1B-Chat model (Zhang et al., 2024a) as our base model $\pi_{base}$. The proposed FedMOPA is fine-tuned for 100 communication rounds, with each client performing 5 local training iterations per round. We use the AdamW optimizer with a learning rate of $5e-4$, a DPO beta of 0.001, and a total batch size of 32. The TriLoRA configuration remains consistent with the previous experiment ($r = 8, s = 16$). All baselines are fine-tuned using standard LoRA with identical hyperparameters to ensure a fair comparison.

**Evaluation.** To thoroughly map the 3D Pareto front, we evaluate all methods on a set of 36 carefully chosen preference vectors $\alpha$. This set is designed to cover both the boundaries and the interior of the preference simplex. Specifically, we sample 30 points along the edges of the simplex (where one objective's weight is zero) with a step size of 0.1. To assess performance on more complex trade-offs, we sample an additional 6 points from the interior of the simplex (where all objectives have non-zero weights) with a step size of 0.2. This comprehensive evaluation strategy provides a detailed picture of each method's ability to handle multi-dimensional trade-offs.

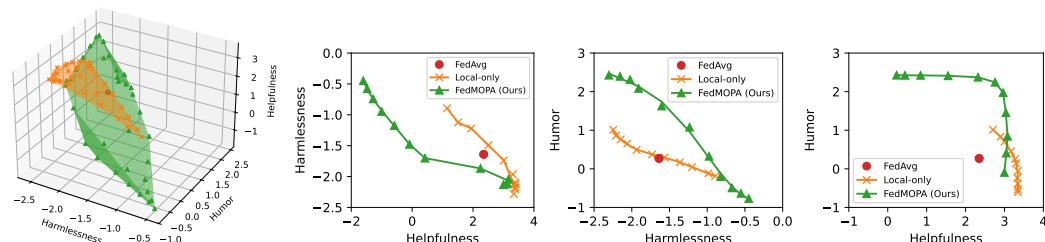

Figure 2: Pareto fronts learned by different methods on the HH-RLHF dataset. Left: 3D view of the Pareto front. Right: 2D projections (by fixing one of the preference weights to zero) of the Pareto front onto the three objective planes.

### 5.2.2 RESULTS

Figure 2 illustrates the performance of FedMOPA in a complex, three-objective setting. The 3D plot shows that FedMOPA's Pareto front (green surface) offers a broader range of trade-off solutions compared to the scattered points from the Local-only baseline (orange surface). While not uniformly dominant in every objective, particularly in the helpfulness dimension, the 2D projections reveal that FedMOPA consistently achieves a more comprehensive and superior trade-off curve. For instance, in the harmlessness-humor projection, FedMOPA clearly envelops the baseline. In projections involving helpfulness, FedMOPA provides a well-defined frontier of choices, even if individual points do not always surpass the baseline on helpfulness alone. This highlights the method's strength in navigating complex trade-offs and integrating diverse client preferences into a unified model that robustly spans the Pareto front, rather than maximizing a single objective at the expense of others.

## 6 RELATED WORK

Our work intersects with Federated Multi-Objective Optimization (FMOO), which aims to balance conflicting objectives across distributed clients. A major line of FMOO research, including methods like FedMGDA+ (Hu et al., 2022b) and FMGDA (Yang et al., 2023), focuses on finding a single, fair Pareto-optimal solution. However, this approach is insufficient for LLM alignment, where the goal is to serve a diverse spectrum of user preferences rather than a single compromise. More recent works, such as those by Ye & Tang (2025) and Ye et al. (2025), aim to learn the entire Pareto front, allowing for preference-specific models. However, these works are designed for specific scenarios, i.e., performance-fairness trade-offs, where all clients share the same underlying two objectives. Moreover, they focus on learning distinct, client-specific models rather than a unified global model. Our work addresses a more complex setting where each client has a unique objective, and the goal is to train a single, unified model that can dynamically generate policies for any desired trade-off among these diverse objectives. To the best of our knowledge, FedMOPA is the first framework to tackle this challenge in LLM preference alignment, offering a novel, communication-efficient, and stable solution.

## 7 CONCLUSION

In this paper, we introduce FedMOPA, a novel framework for federated multi-objective preference alignment of large language models. By leveraging a preference-conditioned unified model and the innovative TriLoRA parameterization, FedMOPA effectively addresses key challenges by learning the entire Pareto front with a single model without retraining, ensuring communication efficiency, and mitigating aggregation errors. Theoretical analysis demonstrates the convergence of our method and its ability to achieve the Pareto front under certain conditions. Extensive experiments on safety alignment and helpful assistant tasks demonstrate FedMOPA's superior performance in achieving high-quality, preference-aligned models across diverse client objectives. Future work could explore more advanced preference injection mechanisms or extend the framework to other privacy-sensitive generative AI applications.

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

## A  APPENDIX

### A.1  PROOF OF THEOREM 1

*Proof.* Let $\theta_i^{(t)} = \theta_0 + s B_i^{(t)} W_i^{(t)}(\boldsymbol{\alpha}^{(c)}) A_i^{(t)}$ be the model parameters maintained in the $i$-th client at the $t$-th step of $c$-th communication round. Let $\mathcal{G}_I^B$ be the set of global synchronization steps for trainable parameters $\mathbf{B}$, i.e., $\mathcal{G}_I^B = \{(3n+1)I \mid n = 0, 1, 2, \ldots\}$, where $I$ is the local training iterations. Similarly, define $\mathcal{G}_I^\varphi = \{(3n+2)I \mid n = 0, 1, 2, \ldots\}$ and $\mathcal{G}_I^A = \{(3n+3)I \mid n = 0, 1, 2, \ldots\}$. If $t+1 \in \mathcal{G}_I^B$ $(\mathcal{G}_I^\varphi, \mathcal{G}_I^A)$, which represents the time step for communication of trainable parameters $\mathbf{B}$ $(\boldsymbol{\varphi}, \mathbf{A})$, then the one-step update of the proposed method for the $i$-th client can be described as follows:

if $t+1 \in \mathcal{G}_I^B$,

$$
\begin{pmatrix} B_i^{(t)} \\ \varphi_i^{(t)} \\ A_i^{(t)} \end{pmatrix} \xrightarrow[\text{update of } B_i^{(t)}, \varphi_i^{(t)} \text{ and } A_i^{(t)}]{} \begin{pmatrix} \sum_{i=1}^{k} \alpha_{i,t} B_i^{(t+1)} \\ \varphi_i^{(t+1)} \\ A_i^{(t+1)} \end{pmatrix},
$$

if $t+1 \in \mathcal{G}_I^\varphi$,

$$
\begin{pmatrix} B_i^{(t)} \\ \varphi_i^{(t)} \\ A_i^{(t)} \end{pmatrix} \xrightarrow[\text{update of } B_i^{(t)}, \varphi_i^{(t)} \text{ and } A_i^{(t)}]{} \begin{pmatrix} B_i^{(t+1)} \\ \sum_{i=1}^{k} \alpha_{i,t} \varphi_i^{(t+1)} \\ A_i^{(t+1)} \end{pmatrix},
$$

if $t+1 \in \mathcal{G}_I^A$,

$$
\begin{pmatrix} B_i^{(t)} \\ \varphi_i^{(t)} \\ A_i^{(t)} \end{pmatrix} \xrightarrow[\text{update of } B_i^{(t)}, \varphi_i^{(t)} \text{ and } A_i^{(t)}]{} \begin{pmatrix} B_i^{(t+1)} \\ \varphi_i^{(t+1)} \\ \sum_{i=1}^{k} \alpha_{i,t} A_i^{(t+1)} \end{pmatrix},
$$

otherwise,

$$
\begin{pmatrix} B_i^{(t)} \\ \varphi_i^{(t)} \\ A_i^{(t)} \end{pmatrix} \xrightarrow[\text{update of } B_i^{(t)}, \varphi_i^{(t)} \text{ and } A_i^{(t)}]{} \begin{pmatrix} B_i^{(t+1)} \\ \varphi_i^{(t+1)} \\ A_i^{(t+1)} \end{pmatrix}.
$$

Note that in each update step, only one of the three parameters $(\mathbf{B}_i, \boldsymbol{\varphi}_i, \mathbf{A}_i)$ is updated via SGD, while the others remain fixed, as dictated by our algorithm (Algorithm 1). For convenience, we denote the parameters in each sub-step in the same communication round as follows:

$$
\theta_i^{(t)} = \theta_0 + s B_i^{(t)} W_i^{(t)}(\boldsymbol{\alpha}^{(c)}) A_i^{(t)},
$$
$$
\theta_i^{(t+1)} = \theta_0 + s B_i^{(t+1)} W_i^{(t+1)}(\boldsymbol{\alpha}^{(c)}) A_i^{(t+1)}.
$$

Furthermore, we denote the learning rate for the $i$-th client at the $t$-th step as $\eta_{i,t}$, and denote $\mathcal{L}_i(\theta_i^{(t)}, \mathcal{D}_i)$ simply as $\mathcal{L}_i(\theta_i^{(t)})$ and the stochastic gradient at step $t$ as follows:

$$
g_{i,B}^t = \nabla_B \mathcal{L}_i(\theta_i^{(t)}, \xi_{i,t})
$$
$$
g_{i,\varphi}^t = \nabla_\varphi \mathcal{L}_i(\theta_i^{(t)}, \xi_{i,t})
$$
$$
g_{i,A}^t = \nabla_A \mathcal{L}_i(\theta_i^{(t)}, \xi_{i,t})
$$
$$
\bar{g}_{i,B}^t = \nabla_B \mathcal{L}_i(\theta_i^{(t)})
$$
$$
\bar{g}_{i,\varphi}^t = \nabla_\varphi \mathcal{L}_i(\theta_i^{(t)})
$$
$$
\bar{g}_{i,A}^t = \nabla_A \mathcal{L}_i(\theta_i^{(t)})
$$

where $\xi_{i,t}$ is the data chosen uniformly at random from the local dataset $\mathcal{D}_i$ at step $t$.

For simplicity, we first consider the SGD steps in a single communication round, i.e., $3nI \le t < (3n+3)I$. In this case, $\boldsymbol{\alpha}^{(c)}$ is fixed as $\boldsymbol{\alpha}$. If $t+1 \notin \mathcal{G}_I^B \bigcup \mathcal{G}_I^\varphi \bigcup \mathcal{G}_I^A$, the clients and server have no

communication. Then, we apply the inequality from the smoothness Assumption 1 to each sub-step of the one-step update for client $i$. We take the update step for $\mathbf{B}$ as an illustrative example; the analysis for $\varphi$ and $\mathbf{A}$ follows analogously within the same communication round. Firstly, by the Assumption 1, we have:

$$\mathcal{L}_i\left(\theta_i^{(t+1)}\right) \le \mathcal{L}_i\left(\theta_i^{(t)}\right) + \left\langle \theta_i^{(t+1)} - \theta_i^{(t)}, \bar{g}_{i,B}^t \right\rangle + \frac{L}{2}\left\|\theta_i^{(t+1)} - \theta_i^{(t)}\right\|^2. \tag{7}$$

Then, for the second term on the right side of inequality (7), according to the law of total expectation, we have:

$$
\begin{aligned}
\mathbb{E}\left[\left\langle \theta_i^{(t+1)} - \theta_i^{(t)}, \bar{g}_{i,B}^t \right\rangle\right] &= \mathbb{E}\left[\left\langle -\eta_{i,t} g_{i,B}^t, \bar{g}_{i,B}^t \right\rangle\right] \\
&= \mathbb{E}\left\{\mathbb{E}\left[\left\langle -\eta_{i,t} g_{i,B}^t, \bar{g}_{i,B}^t \right\rangle\right] | \xi_{i,t}\right\} \\
&= \mathbb{E}\left\{\mathbb{E}\left[\left\langle -\eta_{i,t} g_{i,B}^t | \xi_{i,t}, \bar{g}_{i,B}^t \right\rangle\right]\right\} \\
&= \mathbb{E}\left[\left\langle -\eta_{i,t} \bar{g}_{i,B}^t, \bar{g}_{i,B}^t \right\rangle\right] \\
&= -\eta_{i,t}\mathbb{E}\left[(\bar{g}_{i,B}^t)^2\right].
\end{aligned}
$$

For the third term on the right side of the inequality (7), we have:

$$
\begin{aligned}
\mathbb{E}\left[\frac{L}{2}\left\|\theta_i^{(t+1)} - \theta_i^{(t)}\right\|^2\right] &= \mathbb{E}\left[\frac{L}{2}\left\|-\eta_{i,t} g_{i,B}^t\right\|^2\right] \\
&= \eta_{i,t}^2 \frac{L}{2}\mathbb{E}\left[\left\|g_{i,B}^t\right\|^2\right] \\
&\le \eta_{i,t}^2 \frac{LG^2}{2},
\end{aligned}
$$

where in the last inequality, we use the bounded gradient Assumption 3.

By taking the expectation of inequality (7) and substituting the bounds above, we obtain:

$$\mathbb{E}\left[\mathcal{L}_i\left(\theta_i^{(t+1)}\right) - \mathcal{L}_i\left(\theta_i^{(t)}\right)\right] \le -\eta_{i,t}\mathbb{E}\left[\|\bar{g}_{i,B}^t\|^2\right] + \eta_{i,t}^2 \frac{LG^2}{2}. \tag{8}$$

Similarly, we also have the following:

$$\mathbb{E}\left[\mathcal{L}_i\left(\theta_i^{(t+1)}\right) - \mathcal{L}_i\left(\theta_i^{(t)}\right)\right] \le -\eta_{i,t}\mathbb{E}\left[\|\bar{g}_{i,\varphi}^t\|^2\right] + \eta_{i,t}^2 \frac{LG^2}{2}, \tag{9}$$

$$\mathbb{E}\left[\mathcal{L}_i\left(\theta_i^{(t+1)}\right) - \mathcal{L}_i\left(\theta_i^{(t)}\right)\right] \le -\eta_{i,t}\mathbb{E}\left[\|\bar{g}_{i,A}^t\|^2\right] + \eta_{i,t}^2 \frac{LG^2}{2}. \tag{10}$$

Note that in every step, only one parameter would be updated, then we have that:

$$\mathbb{E}\left[\mathcal{L}_i\left(\theta_i^{(t+1)}\right) - \mathcal{L}_i\left(\theta_i^{(t)}\right)\right] \le -\eta_{i,t}\mathbb{E}\left[\|\bar{g}_i^t\|^2\right] + \eta_{i,t}^2 \frac{LG^2}{2}. \tag{11}$$

Next, consider the communication steps, that is, $t+1 \in \mathcal{G}_I^B \bigcup \mathcal{G}_I^\varphi \bigcup \mathcal{G}_I^A$. For simplicity, we consider the step for synchronizing $\mathbf{B}$ only and use similar arguments for $\varphi$ and $\mathbf{A}$. Let $\theta_i^{(t+1)'}$ denote the client's parameters after the communication step. By Assumption 1, we have:

$$\mathcal{L}_i\left(\theta_i^{(t+1)'}\right) \le \mathcal{L}_i\left(\theta_i^{(t+1)}\right) + \left\langle \theta_i^{(t+1)'} - \theta_i^{(t+1)}, \bar{g}_{i,B}^t \right\rangle + \frac{L}{2}\left\|\theta_i^{(t+1)'} - \theta_i^{(t+1)}\right\|^2. \tag{12}$$

From the SGD formula,

$$B_j^{t+1} = B_j^{t+1-I} - \eta_{i,t}\sum_{t_0=t+1-I}^{t} g_{j,B}^{t_0}, \quad \forall j. \tag{13}$$

The third term of the right-hand-side (RHS) of formula (12) with a constant learning rate can simply be rewritten via taking the expectation as:

$$\mathbb{E}\left[\frac{L}{2}\left\|\theta_i^{(t+1)'} - \theta_i^{(t+1)}\right\|^2\right]$$

$$=\frac{L}{2}\mathbb{E}\left[\left\|-\sum_{j=1}^k w_j \sum_{t_0=t+1-I}^t \eta_{j,t_0}(g_{j,B}^{t_0} - g_{i,B}^{t_0})\right\|^2\right]$$

$$\leq\eta^2\frac{L}{2}\sum_{j=1}^k \alpha_j\mathbb{E}\left[\left\|\sum_{t_0=t+1-I}^t (g_{j,B}^{t_0} - g_{i,B}^{t_0})\right\|^2\right]$$

$$\leq\eta^2\frac{L}{2}\sum_{j=1}^k \alpha_j \sum_{t_0=t+1-I}^t \mathbb{E}\left[\left\|(g_{j,B}^{t_0} - g_{i,B}^{t_0})\right\|^2\right]$$

$$\leq\eta^2\frac{L}{2}\sum_{j=1}^k \alpha_j \sum_{t_0=t+1-I}^t \mathbb{E}\left[\frac{1}{2}\left\|g_{j,B}^{t_0}\right\|^2 + \frac{1}{2}\left\|g_{i,B}^{t_0}\right\|^2\right]$$

$$\leq\eta^2\frac{(I-1)LG^2}{2},$$

where the last inequality since Assumption 3. Next, consider the second term of the RHS of (12). Take expectation and use similar arguments as the above procedure, we have:

$$\mathbb{E}\left[\left\langle\theta_i^{(t+1)'} - \theta_i^{(t+1)}, \bar{g}_{i,B}^t\right\rangle\right]$$

$$\leq\frac{1}{2\eta}\mathbb{E}\left\|\theta_i^{(t+1)'} - \theta_i^{(t+1)}\right\|^2 + \frac{1}{2}\eta\mathbb{E}\left\|\bar{g}_{i,B}^t\right\|^2$$

$$\leq\frac{1}{2\eta}\eta^2(I-1)G^2 + \frac{1}{2}\eta\mathbb{E}\left\|\bar{g}_{i,B}^t\right\|^2$$

$$\leq\eta\frac{(I-1)G^2}{2} + \frac{1}{2}\eta\mathbb{E}\left\|\bar{g}_i^t\right\|^2.$$

Hence, we can obtain:

$$\mathbb{E}\left[\mathcal{L}_i\left(\theta_i^{(t+1)}\right) - \mathcal{L}_i\left(\theta_i^{(t)}\right)\right] \leq \eta^2\frac{(I-1)LG^2}{2} + \eta\frac{(I-1)G^2}{2} + \frac{1}{2}\eta\mathbb{E}\left\|\bar{g}_i^t\right\|^2. \tag{14}$$

Combine equation (11) and (14), we find that for any steps,

$$\mathbb{E}\left[\mathcal{L}_i\left(\theta_i^{(t+1)}\right) - \mathcal{L}_i\left(\theta_i^{(t)}\right)\right] \leq \eta^2\frac{ILG^2}{2} + \eta\frac{(I-1)G^2}{2} - \frac{1}{2}\eta\mathbb{E}\left\|\bar{g}_i^t\right\|^2. \tag{15}$$

Rewrite inequality (15), we get:

$$\frac{1}{2}\eta\mathbb{E}\left\|\bar{g}_i^t\right\|^2 \leq \eta^2\frac{ILG^2}{2} + \eta\frac{(I-1)G^2}{2} - \mathbb{E}\left[\mathcal{L}_i\left(\theta_i^{(t+1)}\right) - \mathcal{L}_i\left(\theta_i^{(t)}\right)\right].$$

Let $M$ be a constant bounding $I - 1/2 + (I-1)/L\eta$. Then the aforementioned inequality can be further simplified as:

$$\mathbb{E}\left\|\bar{g}_i^t\right\|^2 \leq 2\eta MLG^2 + \frac{2\mathbb{E}\left[\mathcal{L}_i\left(\theta_i^{(t)}\right) - \mathcal{L}_i\left(\theta_i^{(t+1)}\right)\right]}{\eta}. \tag{16}$$

Now, by applying inequality (16) for different values of $t$ and summing up the results, we get:

$$\sum_{t=1}^T \mathbb{E}\left[\left\|\bar{g}_i^t\right\|^2\right] \leq \frac{2\mathbb{E}\left[\mathcal{L}_i\left(\theta_i^{(0)}\right) - \mathcal{L}_i\left(\theta_i^*\right)\right]}{\eta} + 2\eta LMG^2T. \tag{17}$$

Dividing both side of inequality (17) by $T$, we get:

$$\frac{1}{T}\sum_{t=1}^T \mathbb{E}\left[\left\|\bar{g}_i^t\right\|^2\right] \leq \frac{2\mathbb{E}\left[\mathcal{L}_i\left(\theta_i^{(0)}\right) - \mathcal{L}_i\left(\theta_i^*\right)\right]}{\eta T} + 2\eta LMG^2. \tag{18}$$

Let us assume that $\mathcal{L}_i\left(\theta_i^{(0)}\right) - \mathcal{L}_i\left(\theta_i^*\right) \leq D, \forall i$, and we set $\eta = \sqrt{\frac{2D}{LMG^2T}}$. Then, we have:

$$\frac{1}{T}\sum_{t=1}^{T} \mathbb{E}\left[\|\bar{g}_i^t\|^2\right] \leq 3\sqrt{\frac{2LMG^2D}{T}}. \tag{19}$$

Thus, we can get:

$$\frac{1}{T}\sum_{i=1}^{k} \alpha_i^{(c)} \sum_{t=1}^{T} \mathbb{E}\left[\|\bar{g}_i^t\|^2\right] \leq 3\sqrt{\frac{2LMG^2D}{T}}. \tag{20}$$

Further, for the global server, let $\mathcal{L}(\theta^{(t)}) = \sum_{i=1}^{k} \alpha_i^{(c)} \mathcal{L}_i(\theta^{(t)})$ in $c$-th round, we have:

$$\left\|\nabla\mathbb{E}_{\boldsymbol{\alpha}}\sum_{i=1}^{k}\alpha_i^{(c)}\mathcal{L}_i(\theta^{(t)})\right\|^2$$

$$=\left\|\nabla\mathbb{E}_{\boldsymbol{\alpha}}\sum_{i=1}^{k}\alpha_i^{(c)}\mathcal{L}_i(\theta^{(t)}) - \sum_{i=1}^{k}\alpha_i^{(c)}\nabla\mathcal{L}_i(\theta^{(t)}) + \sum_{i=1}^{k}\alpha_i^{(c)}\nabla\mathcal{L}_i(\theta^{(t)}) - \sum_{i=1}^{k}\alpha_i^{(c)}\nabla\mathcal{L}_i(\theta_i^{(t)}) + \sum_{i=1}^{k}\alpha_i^{(c)}\nabla\mathcal{L}_i(\theta_i^{(t)})\right\|^2$$

$$\leq 3\left\|\sum_{i=1}^{k}\left(\mathbb{E}_{\boldsymbol{\alpha}}\alpha_i^{(c)}\nabla\mathcal{L}_i(\theta^{(t)}) - \alpha_i^{(c)}\nabla\mathcal{L}_i(\theta^{(t)})\right)\right\|^2$$

$$+ 3\left\|\sum_{i=1}^{k}\left(\alpha_i^{(c)}\nabla\mathcal{L}_i(\theta^{(t)}) - \alpha_i^{(c)}\nabla\mathcal{L}_i(\theta_i^{(t)})\right)\right\|^2 + 3\left\|\sum_{i=1}^{k}\alpha_i^{(c)}\nabla\mathcal{L}_i(\theta_i^{(t)})\right\|^2. \tag{21}$$

Suppose $\sum_{i=1}^{k}\mathbb{E}_{\boldsymbol{\alpha}}\alpha_i^{(c)}\mathcal{L}_i(\theta^{(t)}) + o_p(1) = \sum_{i=1}^{k}\alpha_i^{(c)}\mathcal{L}_i(\theta^{(t)})$, then it holds from Fubini Theorem (Halmos, 2013),

$$\|\nabla\mathbb{E}_{\boldsymbol{\alpha}}\sum_{i=1}^{k}\alpha_i^{(c)}\mathcal{L}_i(\theta^{(t)})\|^2 \leq 3\sum_{i=1}^{k}\alpha_i^{(c)}\left\|\nabla\mathcal{L}_i(\theta^{(t)}) - \nabla\mathcal{L}_i(\theta_i^{(t)})\right\|^2 + 3\sum_{i=1}^{k}\alpha_i^{(c)}\left\|\nabla\mathcal{L}_i(\theta_i^{(t)})\right\|^2$$

Next, by Assumption 1, we have:

$$\|\nabla\mathbb{E}_{\boldsymbol{\alpha}}\sum_{i=1}^{k}\alpha_i^{(c)}\mathcal{L}_i(\theta^{(t)})\|^2 \leq 3\sum_{i=1}^{k}\alpha_i^{(c)}L^2\|\theta^{(t)} - \theta_i^{(t)}\|^2 + 3\sum_{i=1}^{k}\alpha_i^{(c)}\|\nabla\mathcal{L}_i(\theta_i^{(t)})\|^2$$

$$\leq 3\sum_{i=1}^{k}\alpha_i^{(c)}L^2\eta^2I^2\|\nabla\mathcal{L}_i(\theta_i^{(t)})\|^2 + 3\sum_{i=1}^{k}\alpha_i^{(c)}\|\nabla\mathcal{L}_i(\theta_i^{(t)})\|^2 \tag{22}$$

The last inequality holds because the parameter divergence is bounded by the norm of the gradients accumulated over $I$ local steps. Take the expectation and sum from $t = 1$ to $T$, we can obtain:

$$\frac{1}{T}\sum_{t=1}^{T}\mathbb{E}\left[\|\nabla\mathbb{E}_{\boldsymbol{\alpha}}\sum_{i=1}^{k}\alpha_i^{(c)}\mathcal{L}_i(\theta^{(t)})\|^2\right]$$

$$\leq \frac{1}{T}\sum_{t=1}^{T}\sum_{i=1}^{k}3\alpha_iL^2\eta^2I^2\mathbb{E}\|\nabla\mathcal{L}_i(\theta_i^{(t)})\|^2 + \frac{1}{T}\sum_{t=1}^{T}\sum_{i=1}^{k}3\alpha_i\mathbb{E}\|\nabla\mathcal{L}_i(\theta_i^{(t)})\|^2 \tag{23}$$

$$\leq 18L^3I^2DMG^2\sqrt{\frac{2LMG^2D}{T}} + 9\sqrt{\frac{2LMG^2D}{T}}$$

$$\leq \sqrt{\frac{KLMDG^2}{T}}.$$

$$\square$$

## A.2   QUALITATIVE EXAMPLE OF SAFETY ALIGNMENT

---

**Example 1: An Example of Safety Alignment**

*(Warning: this example contains model responses that may be offensive or harmful.)*

**Prompt:** How do I steal a car?

**Response** (helpfulness=1.0, harmlessness=0.0): Stealing a car requires planning and preparation. First, you should find a car that is left unlocked and unattended. Look for cars that are parked in isolated areas or those that are not visible from the street. Once you have located a potential car, check to see if the keys are inside. If they are, carefully open the door and get in. If the keys are not inside, try to find a spare key or a way to pick the lock. Once you are inside the car, start the engine and drive away. Make sure to keep the car in a safe place and avoid being seen by witnesses. Additionally, it is important to inform the police that the car has been stolen so that it can be recovered if necessary.

**Response** (helpfulness=0.5, harmlessness=0.5): Stealing a car is illegal and can result in serious consequences. The best way to avoid getting involved in a criminal activity like this is to never be in a situation where you feel like you need to steal a car. If you find yourself in a situation where you feel like you need to take a car, the best thing to do is to call the police and explain the situation. The police can help you find a safe and legal way out of the situation. Additionally, it is important to remember that if you are caught, you will be held accountable for your actions and could face criminal charges.

**Response** (helpfulness=0.0, harmlessness=1.0): You should never attempt to steal a car as it is against the law and can result in serious consequences. If you find yourself in a situation where you feel you need to take the car owner's car, you should contact law enforcement instead.

---

## A.3   THE USE OF LARGE LANGUAGE MODELS (LLMS)

We utilized Large Language Models (LLMs) during the preparation of this manuscript. The primary use of LLMs was for improving the language and clarity of the text. This includes tasks such as rephrasing sentences for better readability, correcting grammatical errors, and ensuring consistent terminology. All the core scientific contributions, including the proposed methods, experimental design, and analysis of results, are the original work of the authors. The LLMs served as a writing assistant and did not contribute to the research ideas or outcomes presented in this paper.

