# OpenReview forum: "FedMOPA: Federated Multi-Objective Preference Alignment for Large Language Models"
_ICLR.cc/2026/Conference — ICLR 2026 Conference Withdrawn Submission_

### Official Review · Reviewer_EEKu · 2025-10-24

**Soundness:** 2
**Presentation:** 2
**Contribution:** 2
**Rating:** 4
**Confidence:** 4

**Summary:**

This paper proposes FedMOPA to tackle with two problems in aligning LLMs with diverse preferences in federated learning setting.
* To avoid the expensive cost of re-training when adapting to personalized preferences, FedMOPA uses a lightweight hyper-network to generate the parameters of an intermediate layer in LoRA conditioned on the preference vector.
* To avoid the aggregation error of LoRA in federated learning, FedMOPA optimizes LoRA parameters in an alternating strategy.

Experiments are conducted using Alpaca-7B on PKU-SafeRLHF dataset and TinyLlama-1.1B on HH-RLHF dataset.
FedMOPA is compared with FedAvg and post-hoc model merging, where it enjoys superior Pareto front.

**Strengths:**

* This paper has a good structure, making it easy to follow.
* It is interesting to use a hyper-network to generate model parameters to adapt to personalized preferences.

**Weaknesses:**

I deem the analysis and empirical study are not very sufficient, so I lean to reject the paper.
I would like to raise my score if these concerns are well addressed.

* (Minor) I suspect the correctness of Lemma 1.
Consider such an example: $L_1=\theta,L_2=-\theta$.
It is clear that all $\theta\in\mathbb R$ lies on the Pareto front because an increase in $L_1=\theta$ necessarily corresponds to a decrease in $L_2=-\theta$.
Now given $\alpha=(1,0)\in \Delta_1$, following Lemma 1, $\theta$ is Pareto optimal if and only if $\theta$ minimizes $L(\theta|\alpha)=L_1(\theta)=\theta$.
* The paper identifies aggregation error as one of the main challenges.
It would be helpful to introduce the cause of the problem in preliminary and analyze why alternating optimization can address such issue in methodology.
* In Figure 2, when the weight of humor is fixed to zero, the performance of FedMOPA is consistently inferior to FedAvg and post-hoc model merging.
* Ablation study is missing.
For example, if alternating optimization is disabled, to what extend will the model performance degrade?
I see on line 10 of Algorithm 1 preference-weighted FedAvg is used.
If vanilla FedAvg is used instead, will the model performance degrade?

**Questions:**

* The algorithm requires sampling preference vector from simplex.
Does this introduce additional training cost?
How does the training complexity change with respect to the dimension of preferences?
* In the first experiment, metrics HV and MIP are used to demonstrate the superiority of FedMOPA.
Why are these two metrics not used in the second experiment?

---

### Official Review · Reviewer_2kut · 2025-10-31

**Soundness:** 2
**Presentation:** 2
**Contribution:** 2
**Rating:** 2
**Confidence:** 3

**Summary:**

This paper introduces FedMOPA, a framework that integrates federated learning (FL) with multi-objective preference alignment (MOO) for large language models (LLMs). The goal is to train a unified model that can dynamically represent multiple preference trade-offs using a preference vector, without retraining for each preference configuration. To achieve this, the authors propose: TriLoRA, a conditional LoRA module that injects preference information through a conditioning matrix; an alternating optimization strategy to reduce parameter aggregation errors in the federated setting.
The paper provides theoretical convergence analysis and reports experimental improvements over Local-only and FedAvg baselines on two multi-objective LLM alignment tasks (safety and helpfulness).

**Strengths:**

1. Conceptual contribution: Proposing a unified, preference-conditioned model to avoid retraining for each preference configuration is appealing and potentially impactful.
2. Technical components: The introduction of TriLoRA and alternating optimization provides a direction to handle preference-conditioned low-rank updates in FL.

**Weaknesses:**

1. Lack of discussion on the utility and trade-offs of the unified model.
Under equal resources (data, computation, training time), does the unified model perform as well as models trained for specific preferences? If not, how large is the performance gap, and is it justified by flexibility?What is the efficacy vs. efficiency trade-off?
The paper claims to “solve” this challenge but does not analyze the costs in expressiveness or accuracy. This is a major weakness.

2. The paper assumes each local client corresponds to one preference, which is overly idealized in federated settings. $\alpha$ is both a client weight and a preference weight. In reality, client data often reflect mixed or evolving preferences. While simplifying to single preferences per client may be acceptable for initial work, this limitation should be explicitly stated.

3.  Rationality of the “single model” assumption and $\alpha$-space scalability: FedMOPA proposes a single, preference-conditioned model by adding $\alpha$ as an input variable.However, as the number of clients or objectives increases, the $\alpha$-space becomes high-dimensional and harder to represent.Furthermore, each client has limited local data, which seems inconsistent with the high degree of freedom introduced by $\alpha$.Scalability experiments or theoretical analysis are needed to verify the model’s generalization ability as the number of clients or objectives increases.

4. Inadequate explanation of TriLoRA and $W(\alpha)$ design: TriLoRA and $W(\alpha)$ are the core innovations but are poorly described.

5. Insufficient baseline comparisons: The paper only compares FedMOPA with Local-only and FedAvg, which are weak baselines. Without stronger baselines, it is difficult to assess the true benefits of the proposed method.

**Questions:**

Please refer to the “Weakness” section for related questions.

---

### Official Review · Reviewer_j9wK · 2025-11-01

**Soundness:** 2
**Presentation:** 3
**Contribution:** 2
**Rating:** 4
**Confidence:** 5

**Summary:**

This work studies on privacy-preserved multi-objective preference alignment for LLMs and proposes a new method based on federated learning.

While the topic is important and the paper presents a method that is conceptually reasonable, I have significant concerns on the novelty, unclear motivation and insufficient experiments.
As a result, I assign a score of 4.

**Strengths:**

1.	This work studies on an important problem.
2.	The proposed method appears to be reasonable.
3.	Extensive experiments on real-world datasets have been conducted to verify the efficacy of the proposed method.

**Weaknesses:**

1.	My major concern lies on the novelty. This paper examines multiple aspects of DPO, including privacy preservation, multi-objective optimization, and efficient fine-tuning. However, each of these subproblems has already been extensively studied in prior work. For example, federated alignment has been explored in [a1], [a2]; multi-objective optimization for DPO has been addressed in [a3], [a4]; and numerous variants of LoRA have been thoroughly investigated in [a5]. The paper does not sufficiently review these advances nor clarify how the proposed method differs from or outperforms them. As a result, the contribution appears fragmented, and the method itself comes across as a straightforward “A+B+C” combination rather than a genuine breakthrough.


2.	My another concern lies on the unclear motivation. While this work aims to address all of these subproblems simultaneously, the challenges of doing so are not clearly articulated. The authors should explicitly state the difficulties in tackling these aspects jointly and explain how the proposed approach is tailored to overcome them. At present, the method does not appear closely designed for the problem at hand. For example, in Section 3.2.1, it is unclear what specific challenges TriLoRA is meant to address and why existing LoRA strategies could not handle them.

3.	There are also some concerns on the experiments: 1) Some important baselines like [a1-a4] are missing. 2) The experiments are only conducted on a relatively small LLM (TinyLLama-1.1B). It would be better to demonstrate its effectivenss on larger and diverse LLMs. 3) The experiments are only conducted only on one benchmark. Including more datasets would provide stronger empirical support and generalizability.


[a1] WWW’25: Federated fine-tuning of large language models: Kahneman-Tversky vs. direct preference optimization
[a2] KDD’24: Openfedllm: Training large language models on decentralized private data via federated learning
[a3] AAAI’25: Robust Multi-Objective Preference Alignment with Online DPO
[a4] arxiv’24: Hyperdpo: Conditioned one-shot multi-objective fine-tuning framework
[a5] FCS’24: A survey on lora of large language models

**Questions:**

Please refer to weakneses.

---

### Official Review · Reviewer_B7dG · 2025-11-06

**Soundness:** 2
**Presentation:** 2
**Contribution:** 1
**Rating:** 2
**Confidence:** 3

**Summary:**

The paper proposes FedMOPA to solve multi-objective preference alignment in federetaed learning scenario. To do this, authors design TriLoRA, which is a conditional LoRA to control any preference signal vector during the inference. Experiments are synthetically done with PKU-SafeRLHF-30K dataset with 3 objectives: Harmlessness, Helpfulness, and humor and FedMOPA shows modest performance.

**Strengths:**

* The motivation is clear and it touches important problem of multi-objective alignment in FL setting.
* Theoretical anlaysis on the convergence is done.

**Weaknesses:**

* **Weak experimental results** : While FedMOPA is designed to reach any point in the pareto-frontier, the experimental results does not support this. Specifically the result of helpfulness in 5.2 clearly shows that the certain metrics can even be decreased (Figure 2-(c)).

* **Novelty** : While the proposed method propose TriLoRA which is conditional LoRA that ideally puts any preference vector, this concept is not entirely novel (see [1] for example).

* **Real scenario with more objectives** is needed to argue the effectiveness of the proposed method for multi-objective alignment in FL setting. Synthetically split the dataset with only 3 objectives with only 3 client is far from the realistic setting.

**Questions:**

* Have authors investigated inference-time controlloing method (for example applying CFG-style [2]) or test-time scaling with trained model?

* Can author test the result with another base model?

* Can you provide the hyper-parameter sensitivity of the proposed approach? Specifically, how's learning rate or $\beta$ in DPO affect the overall training?


[1] (Stracke et al.) CTRLorALTer: Conditional LoRAdapter for Efficient 0-Shot Control & Altering of T2I Models.

[2] (Ho et al.) Classifier-Free Diffusion Guidance

---

### Note · Authors · 2025-12-23

I have read and agree with the venue's withdrawal policy on behalf of myself and my co-authors.